# Towards Scalable Distance-Enhanced Graph Neural Network

## Abstract

Graph neural networks (GNNs) have demonstrated significant advantages in graph mining tasks, but often suffer from limited expressive power. Among existing expressive GNNs, distance-enhanced GNNs (DE-GNNs) arise as promising ones due to their conceptual simplicity and alignment with the expressive needs of real-world applications. However, scalability remains a key challenge for DE-GNNs, as constructing pairwise distance features requires quadratic complexity. Additionally, while existing work has shown that specialized distance features enable strong expressiveness, the expressive power of simpler distance metrics remains less understood. In this paper, we propose a new **S**calable **D**istance-**E**nhanced **G**raph **N**eural **N**etwork (termed SDE-GNN) to tackle the above issues. SDE-GNN introduces a distance-aware message-passing framework, where message weights are computed by a learnable distance feature mapping. It first linearly projects the adjacency-power-based distance vector to a scalar, then applies a polynomial expansion. To efficiently scale to large graphs, we reformulate the distance features as the product of two asymmetric node encodings and apply Randomized SVD for dimensionality reduction, lowering the computational complexity from quadratic in the number of nodes to linear in the number of edges. Additionally, we leverage the sparsity of the adjacency matrix to directly compute the first-order term of the distance feature mapping, further mitigating distortion from dimensionality reduction. Theoretically, we show that the adopted adjacency-power-based distance outperforms other commonly used distance features. Empirically, we conduct experiments on 17 datasets and verify the effectiveness, efficiency, and scalability of SDE-GNN.

## 1 Introduction

Graph-structured data, which represents entities and their relationships as nodes and edges, is ubiquitous across domains ranging from social networks to natural sciences. Although Graph Neural Networks (GNNs) (Kipf & Welling, 2017; Hamilton et al., 2017; Velickovic et al., 2018) have achieved remarkable success in graph mining tasks, it is widely acknowledged that standard GNNs have suffered from limited expressive power. For instance, they are constrained by the 1-Weifeiler-Lehman test in their ability to distinguish graph isomorphisms (Xu et al., 2019), and are incapable of counting certain simple substructures (Chen et al., 2020), and fail to distinguish specific node pairs (Zhang & Chen, 2018). These limitations hinder their effectiveness in tasks that demand higher expressive capacity.

Improving the expressiveness of GNNs has been extensively explored in recent years (Morris et al., 2020b; 2019; Cotta et al., 2021; Bevilacqua et al., 2022; Abboud et al., 2021; Zhang et al., 2024; 2023). Among these approaches, distance-enhanced GNNs (DE-GNNs) have attracted remarkable attention due to their conceptual simplicity, which enhances the message-passing procedure by incorporating distance features. Specifically, when node $u$ aggregates features from its neighbor node $v$, DE-GNNs concatenate an additional distance feature $\mathrm{d}(u, v)$ with the original features transmitted from $v$ to $u$. Promising results have been shown on the expressiveness of DE-GNNs, including the capability to distinguish graphs beyond 1-WL (Li et al., 2020), the ability to count specific graph substructures (Ma et al., 2023), the identification of critical nodes and edges (Zhang et al., 2023), and awareness of the affinity between nodes (Velingker et al., 2023), which closely align with the demands of practical applications.

Despite recent progress, research on DE-GNNs remains unsatisfactory. One major limitation lies in scalability. As shown in Table 1, existing methods often require at least quadratic computational complexity to construct pairwise distance features among nodes, making them suitable only for graphs with fewer than a few thousand nodes. Besides, the theoretical understanding of DE-GNNs is also insufficient. Existing quantitative expressiveness analyses only demonstrate strong expressiveness for DE-GNN with specialized distances like eigen projection distance and resistance distance (Zhang et al., 2024; 2023). This naturally leaves a question regarding the expressiveness of DE-GNNs when equipped with more basic distance features, such as the power of the adjacency matrix.

To preserve the expressive power of DE-GNNs while ensuring scalability, we propose a **S**calable **D**istance-**E**nhanced **G**raph **N**eural **N**etwork, termed SDE-GNN. We begin by introducing a distance-aware message-passing framework, where message weights between nodes are modulated by their pairwise distance features, defined as entries from powers of the adjacency matrix. Specifically, we map these features into scalar weights via a learnable distance feature mapping, which performs linear projection followed by a nonlinear polynomial expansion, effectively capturing complex patterns in the distance features. Instead of explicitly computing the distance feature mapping, we reformulate the distance features as the product of two asymmetric node encodings and then perform Randomized SVD (Halko et al., 2011) to compress the asymmetric node encodings, thereby reducing the overall complexity from quadratic in the number of nodes to linear in the number of nodes and edges. Additionally, we leverage the sparsity of the adjacency matrix to directly compute the first-order term of the distance feature mapping, further mitigating distortion from dimensionality reduction. To validate the effectiveness of the proposed method, we first show that the expressive power of the adopted adjacency-power-based distance—a widely used metric—theoretically upper-bounds that of the eigenspace projection distance, which has been previously shown to be more expressive than other commonly used distance features. Then, empirically, we evaluate SDE-GNN against 15 popular baselines on 17 widely used datasets. The experimental results verify the effectiveness, efficiency, and scalability of SDE-GNN. Our contributions can be summarized as follows:

- **Scalable DE-GNN**. By introducing decouplable polynomial distance encoding and adaptive dimensionality reduction mechanisms, we optimize the computational complexity of DE-GNNs from quadratic w.r.t number of nodes into linear w.r.t number of nodes and edges, making it scalable to larger graphs while preserving strong expressive power.

- **Theoretical Analysis**. We theoretically analyze the role of the adopted adjacency-power-based distance features in message passing, and prove that the expressive power of such distance features upper-bounds that of other commonly used ones.

- **Empirical Validation**. Extensive experiments on 17 benchmarked datasets demonstrate that SDE-GNN achieves superior performance compared to 15 popular baselines in terms of effectiveness, efficiency and scalability.

## 2 RELATED WORKS

Expressive graph neural networks have been a key focus in the graph learning community in recent years. Existing methods can broadly divided into high-order GNNs, subgraph GNNs, and positional encoding-enhanced GNNs (Zhang et al., 2024). High-order GNNs (Morris et al., 2020b; 2019; Maron et al., 2019b;a) achieve expressiveness comparable to high-dimensional WL tests by performing message passing between node tuples but are hindered by prohibitive computational and storage costs. Subgraph GNNs (Cotta et al., 2021; Bevilacqua et al., 2022), on the other hand, offer a more practical approach by dividing the original graph into a collection of subgraphs and performing message passing on each. However, these methods often require a large number of subgraphs—typically equal to the number of nodes — which restricts their scalability to small graphs (Bevilacqua et al., 2024). Positional encoding-enhanced GNNs represent a conceptually simpler way to improve expressiveness, as they augment node and edge features with positional encodings without altering the model architecture, which can be roughly classified into absolute positional encodings (APEs) and relative positional encodings (RPEs). APEs, inspired by positional encodings in 1-D sequence models like Transformers, are identifier vectors of nodes to represent their positions within the graph. Examples include random node features (Abboud et al., 2021; Sato et al., 2021) and eigenvectors of the Laplacian matrix (Kreuzer et al., 2021; Dwivedi & Bresson, 2020). However, the absence of a canonical node ordering in graphs often results in ambiguity (Wang et al., 2022) or necessitates

complex preprocessing (Lim et al., 2023; Huang et al., 2024). More naturally for graphs, RPEs capture pairwise relationships between nodes, serving as additional edge features, with examples including random walk probabilities (Ma et al., 2023), shortest path distance (Li et al., 2020), resistance distance (Velingker et al., 2023), and eigenspace projection distance (Zhang et al., 2024). We refer to GNNs equipped with RPEs as distance-enhanced GNNs in this paper. One limitation of DE-GNNs is that constructing pairwise distance features requires quadratic computational complexity, which limits their application to small graphs. To address this limitation, we propose a scalable DE-GNN that exhibits linear complexity and is suitable for large-scale graphs.

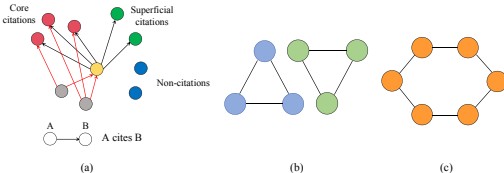

(a)  (b)  (c)

Figure 1: Examples to show the expressiveness of DE-GNNs. (a) depicts a real-world citation network with core citations, superficial citations, and non-citations. (b) and (c) illustrate two regular graphs, where the objective is to detect nodes within triangles.

Table 1: Time complexity comparison of distance-enhanced GNNs for computing all node representations, where $n$, $e$, and $p$ denote the number of nodes, edges, and reduced dimension of our method (with $p \ll n$), respectively.

| Model | Time Complexity |
|---|---|
| DE-GNN (Li et al., 2020) | $O(ne + n^2)$ |
| Graphormer (Ying et al., 2021b) | $O(ne + n^2)$ |
| RD-WL (Zhang et al., 2023) | $O(n^3)$ |
| GRIT (Ma et al., 2023) | $O(ne + n^2)$ |
| EPWL (Zhang et al., 2024) | $O(n^3)$ |
| SDE-GNN (ours) | $O(ep + np^2)$ |

## 3 METHODOLOGY

In this section, we first introduce the overall framework of SDE-GNN in Section 3.1 and then show how to efficiently compute such a framework in Section 3.2. The final computation procedure of SDE-GNN is illustrated in Algorithm 1.

---

**Algorithm 1** The Computation Procedure of SDE-GNN in Matrix Form.

---

**Require:** A graph $\mathcal{G} = (\mathcal{V}, \mathcal{E})$ with its normalized adjacency matrix $\widetilde{A} \in \mathbb{R}^{n \times n}$ and node features $H \in \mathbb{R}^{n \times d}$, reduced embedding dimension $p$, learned parameters $w \in \mathbb{R}^{k+1}, \{\alpha_i\}_{i=1}^m, \theta^{f_m}$.

   ▽ Dimensionality reduction

1: $U \in \mathbb{R}^{n \times p}, \Sigma \in \mathbb{R}^{p \times p}, V \in \mathbb{R}^{n \times p} \leftarrow \text{RandomSVD}(\widetilde{A})$

2: $E \leftarrow \left[ V \parallel U\Sigma \parallel \widetilde{A}U\Sigma \parallel \cdots \parallel \widetilde{A}^{k-1}U\Sigma \right] \in \mathbb{R}^{n \times (k+1) \times p}$

   ▽ Compute the first-order representations

3: $H_1' \leftarrow \alpha_1 \sum_{i=0}^{k} w_i \widetilde{A}^i f_m(H)$

   ▽ Compute higher-order representations

4: $P \leftarrow \sum_{u \in \mathcal{V}} V_u \otimes f_m(H_u) \in \mathbb{R}^{p \times d}$

5: $S \leftarrow \sum_{i=0}^{k} w_i E^{(i)};$   `# Let` $E^{(0)}, \ldots, E^{(k)} \in \mathbb{R}^{n \times p}$ `be the` $k+1$ `matrices forming` $E$

6: $H' \leftarrow H_1' + \sum_{j=2}^{m} \alpha_j (S^{\odot j})P$

7: **Return:** $H'$.

---

### 3.1 DISTANCE-AWARE MESSAGE PASSING FRAMEWORK

**Overall framework.** Motivated by prior work on distance-enhanced GNNs, we introduce a distance-aware message passing mechanism to enhance the expressive power of standard GNNs. Unlike conventional fixed and local schemes (Kipf & Welling, 2017; Rampásek et al., 2022), our framework performs global and adaptive aggregation: each node gathers information from all other nodes rather than its immediate neighbors, with each contribution modulated by a distance-based feature, allowing the model to selectively incorporate relevant signals during message passing. Formally, given a graph $\mathcal{G}(\mathcal{V}, \mathcal{E})$, our message passing framework can be formulated as

$$h_u^{(l+1)} = \sum_{v \in \mathcal{V}} f_e^{(l)}(d_{u,v}) \cdot f_m^{(l)}(h_v^{(l)}), \tag{1}$$

where $\boldsymbol{h}_v^{(l)} \in \mathbb{R}^d$ is the representation of node $v$ at layer $l$, $\boldsymbol{h}_u^{(l+1)} \in \mathbb{R}^d$ is the representation of node $u$ at layer $l+1$, $\boldsymbol{d}_{u,v}$ is the distance features between $u$ and $v$, $f_e^l(\cdot)$ is a distance learning function that maps the distance feature into a scalar to measure the affinity between $u$ and $v$, and $f_m^{(l)}(\cdot)$ is the feature transformation function. For notation simplicity, we omit the superscripts of $(\cdot)^{(l)}$ and replace $(\cdot)^{(l+1)}$ with $(\cdot)'$ in the following statements.

**Detailed configurations.** The three key components of Equation (1) are the distance features $\boldsymbol{d}_{u,v}$, the distance encoding function $f_e(\cdot)$, and the feature transformation function $f_m(\cdot)$. As the primary objective of this paper is to efficiently leverage distance features to enhance GNN expressiveness, we adopt a simple MLP for $f_m(\cdot)$ and focus on the choices of $\boldsymbol{d}_{u,v}$ and $f_e(\cdot)$. For $\boldsymbol{d}_{u,v}$, we adopt a basic choice—the powers of the adjacency matrix. Formally, let $\widetilde{\boldsymbol{A}} = \boldsymbol{D}^{-\frac{1}{2}} \boldsymbol{A} \boldsymbol{D}^{-\frac{1}{2}}$ denote the normalized adjacency matrix, the distance feature is defined as

$$\boldsymbol{d}_{u,v} = [\widetilde{A}_{u,v}^0, \widetilde{A}_{u,v}^1, \cdots, \widetilde{A}_{u,v}^k]^\top \in \mathbb{R}^{k+1}, \tag{2}$$

where $k$ is a hyperparameter to control the power of the adjacency matrix. Here we adopt the power of the normalized adjacency matrix $\tilde{\boldsymbol{A}}^k$ rather than that of the original adjacency matrix (i.e., $\boldsymbol{A}^k$), as the entries of $\boldsymbol{A}^k$ will grow rapidly with $k$. The motivation behind our choice of the adjacency-power-based distance is that it serves as the most basic distance measure in the graph, from which other distances may be derived. For instance, the shortest path distance between $u$ and $v$ is given by the smallest $i$ for which the entry $A_{u,v}^i$ is non-zero. Aligning with such intuition, in Section 4, we will theoretically prove that the expressive power of the adopted adjacency-power-based distance upper-bounds that of the eigenspace projection distance, which (Zhang et al., 2024) has shown to possess the strongest expressive power among other commonly used distances. For the distance learning function $f_e(\cdot)$, we define it as $f_e(\boldsymbol{d}_{u,v}) = \sigma\left(\boldsymbol{w}^\top \boldsymbol{d}_{u,v}\right)$, where $\boldsymbol{w} \in \mathbb{R}^{k+1}$ is a learnable weight vector that maps the $(k+1)$-dimensional adjacency-power-based distance vector to a scalar, bias term is omitted for brevity, and $\sigma$ is an nonlinear function. Such a design of $f_e(\boldsymbol{d}_{u,v})$ enables the adaptive utilization of distinct components in distance features, thereby allowing the model to selectively integrate relevant signals during the message passing process. To characterize the nonlinear function in $f_e(\boldsymbol{d}_{u,v})$, we introduce a polynomial expansion scheme grounded in the Weierstrass approximation theorem, which guarantees that any continuous function defined on a closed interval can be uniformly approximated by polynomials to arbitrary precision (Pérez & Quintana, 2008). Specifically, we redefine $f_e(\boldsymbol{d}_{u,v})$ with a learnable polynomial expansion as

$$f_e(\boldsymbol{d}_{u,v}) = \sum_{i=1}^m \alpha_i \left(\boldsymbol{w}^\top \boldsymbol{d}_{u,v}\right)^i, \tag{3}$$

where $m$ is hyperparameter that denotes the maximum order of polynomial expansion, and $\{\alpha_i\}_{i=1}^m$ are learnable parameters to flexibly control the coefficients for each polynomial term.

**Illustration of expressive advantages.** In Figure 1, we present illustrative examples to demonstrate the expressive advantage of the distance features in capturing nuanced structural relationships, illustrated by the example of triangle detection. Specifically, Figure 1 (b) and (c) illustrate two graphs where vanilla GNNs (Kipf & Welling, 2017) fail to determine whether a node is part of a triangle due to identical local structures. In contrast, the proposed distance-enhanced method can distinguish such cases by identifying whether there exists another node that is both one-hop and two-hop away—an indicator of triangle membership and can be determined by the distance features between nodes. Figure 1 (a) further demonstrates how this triangle-detection capability can be used to differentiate important and less important neighbors in a real-world citation network, where each node represents a paper. In this example, the yellow node makes two types of citations: core citations (e.g., the red node, which forms a triangle with the yellow node) and superficial citations (which do not form such triangles). Due to space limitations, we leave more details of Figure 1 in Appendix A.

## 3.2 EFFICIENT IMPLEMENTATION OF DISTANCE-AWARE MESSAGE PASSING FRAMEWORK

One limitation of the distance feature mapping in Equation (3) is the requirement of explicitly computing the pairwise distances between all node pairs, which incurs a quadratic time complexity. This limits its scalability to large-scale graphs. To address this, we propose an efficient implementation of distance-aware message passing framework, including decouplable message passing framework and efficient dimensionality reduction mechanism.

**Decouplable message passing framework.** We first represent each node by two asymmetric encodings $\boldsymbol{E}_u$ and $\boldsymbol{e}_v$ derived from the adjacency-power-based distance features as

$$\boldsymbol{E}_u = \left[ \widetilde{\boldsymbol{A}}_{u,:}^0 \parallel \widetilde{\boldsymbol{A}}_{u,:}^1 \parallel \cdots \parallel \widetilde{\boldsymbol{A}}_{u,:}^k \right], \boldsymbol{e}_u = \widetilde{\boldsymbol{A}}_{u,:}^0, \tag{4}$$

where $\boldsymbol{E}_u \in \mathbb{R}^{n \times (k+1)}$ comprises the distance features from node $u$ to all other nodes, $\boldsymbol{e}_u \in \mathbb{R}^n$ is a unit vector extracted from $\widetilde{\boldsymbol{A}}^0$, and $\parallel$ denotes the concatenation operation. Since $\boldsymbol{d}_{u,v} = \boldsymbol{E}_u^\top \boldsymbol{e}_v$, we reformulate the distance feature mapping in Equation (3) and decouple it as

$$f_e(\boldsymbol{d}_{u,v}) = \sum_{i=1}^m \alpha_i \left( \boldsymbol{w}^\top \boldsymbol{E}_u^\top \boldsymbol{e}_v \right)^i = \sum_{i=1}^m \alpha_i \left( \boldsymbol{w}^\top \boldsymbol{E}_u^\top \right)^{\odot i} \boldsymbol{e}_v, \tag{5}$$

where $(\cdot)^{\odot i}$ denotes the Hadamard power of order $i$. The polynomial is decouplable as $\boldsymbol{e}_v$ is a one-hot vector. Then the overall message passing framework can be formalized as

$$\boldsymbol{h}_u' = \sum_{v \in \mathcal{V}} f_e(\boldsymbol{d}_{u,v}) \cdot f_m(\boldsymbol{h}_v) = \sum_{i=1}^m \alpha_i (\boldsymbol{w}^\top \boldsymbol{E}_u^\top)^{\odot i} \cdot \sum_{v \in \mathcal{V}} \boldsymbol{e}_v \otimes f_m(\boldsymbol{h}_v), \tag{6}$$

where $\otimes$ denotes the outer product of two vectors.

**Efficient dimensionality reduction mechanism.** To reduce the computational complexity of the message passing framework from quadratic to linear, we propose an efficient dimensionality reduction mechanism without compromising performance. Building upon the theoretical foundation that Singular Value Decomposition (SVD) provides the optimal low-rank approximation of a matrix in both Frobenius and spectral norms (Eckart & Young, 1936), we adopt its faster variant—Randomized SVD (Halko et al., 2011), to decompose the normalized adjacency matrix $\widetilde{\boldsymbol{A}}$ for dimensionality reduction. Specifically, $\widetilde{\boldsymbol{A}}$ is decomposed as $\widetilde{\boldsymbol{A}} \approx \boldsymbol{U}\boldsymbol{\Sigma}\boldsymbol{V}^\top$, where $\boldsymbol{U} \in \mathbb{R}^{n \times p}$, $\boldsymbol{\Sigma} \in \mathbb{R}^{p \times p}$, and $\boldsymbol{V} \in \mathbb{R}^{n \times p}$ are the factor matrices. The reduced dimensionality $p$ is empirically set to $1,000$ to strike a balance between efficiency and efficacy. Then the two asymmetric representations $\boldsymbol{E}_u$ and $\boldsymbol{e}_u$ can be redefined, and we take their matrix form as

$$\boldsymbol{E} = \left[ \boldsymbol{V} \parallel \boldsymbol{U}\boldsymbol{\Sigma} \parallel \widetilde{\boldsymbol{A}}\boldsymbol{U}\boldsymbol{\Sigma} \parallel \cdots \parallel \widetilde{\boldsymbol{A}}^{k-1}\boldsymbol{U}\boldsymbol{\Sigma} \right], \boldsymbol{e} = \boldsymbol{V}, \tag{7}$$

where $\boldsymbol{E} \in \mathbb{R}^{n \times k \times p}$ and $\boldsymbol{e} \in \mathbb{R}^{n \times p}$. A key efficient computational technique is avoiding explicit $n \times n$ matrix construction by iteratively computing $\widetilde{\boldsymbol{A}}^{k-1}\boldsymbol{U}\boldsymbol{\Sigma}$ from right to left—starting with $\boldsymbol{U}\boldsymbol{\Sigma}$, then $\widetilde{\boldsymbol{A}}\boldsymbol{U}\boldsymbol{\Sigma}$, $\widetilde{\boldsymbol{A}}^2\boldsymbol{U}\boldsymbol{\Sigma}$, till $\widetilde{\boldsymbol{A}}^{k-1}\boldsymbol{U}\boldsymbol{\Sigma}$—ensuring all intermediate results remain $n \times p$. Leveraging the sparsity of $\widetilde{\boldsymbol{A}}$, the computational complexity of $\widetilde{\boldsymbol{A}}^{k-1}\boldsymbol{U}\boldsymbol{\Sigma}$ is linear in $\mathrm{nnz}(\widetilde{\boldsymbol{A}})$, which corresponds to the number of edges $e$. Consequently, the complexity of constructing $\boldsymbol{E}$ and $\boldsymbol{e}$ is linear in $e$, and the total computational complexity of $\{\boldsymbol{h}_u'\}_{u \in \mathcal{V}}$ in Equation (6) is linear in $e$.

**Separation of first- and higher-order polynomial terms.** The dimensionality reduction of $\widetilde{\boldsymbol{A}}$ inevitably incurs information loss, which may degrade model performance. To mitigate the information loss, we propose a mechanism that separates the first- and higher-order terms in the polynomial expansion, preserving the first-order term without dimensionality reduction. Specifically, the first-order term without dimensionality reduction in Equation (6) can be reformulated as:

$$\boldsymbol{h}_{u,1}' = \sum_{v \in \mathcal{V}} \alpha_1 (\boldsymbol{w}^\top \boldsymbol{E}_u^\top \boldsymbol{e}_v) \cdot f_m(\boldsymbol{h}_v) = \alpha_1 \sum_{i=0}^k w_i \sum_{v=1}^n \widetilde{\boldsymbol{A}}_{u,v}^i f_m(\boldsymbol{h}_v), \tag{8}$$

where $w_i$ denotes the $i$-th entry of $\boldsymbol{w}$ and $n$ denotes the number of nodes. By stacking the initial node features $\{\boldsymbol{h}_u\}_{u \in \mathcal{V}}$ into a matrix $\boldsymbol{H} \in \mathbb{R}^{n \times d}$, Equation (8) can be rewritten in matrix form as $\boldsymbol{H}_1' = \alpha_1 \sum_{i=0}^k w_i \widetilde{\boldsymbol{A}}^i f_m(\boldsymbol{H})$, where $f_m(\boldsymbol{H})$ denotes row-wise application of $f_m(\cdot)$, and $\boldsymbol{H}_1' \in \mathbb{R}^{n \times d}$ represents the first-order node representations. Notably, this computation can be efficiently performed by exploiting the sparsity of $\widetilde{\boldsymbol{A}}$, similar to Equation (7). Through this separation mechanism, we reformulate the message passing procedure in Equation (6) into a matrix form as

$$\boldsymbol{h}_u' = \boldsymbol{h}_{u,1}' + \sum_{j=2}^m \alpha_j (\boldsymbol{w}^\top \boldsymbol{E}_u^\top)^{\odot j} \cdot \sum_{v \in \mathcal{V}} \boldsymbol{e}_v \otimes f_m(\boldsymbol{h}_v) \tag{9}$$

**Time complexity analysis.** We analyse the time complexity of SDE-GNN and compare it with other distance-enhanced GNNs. The time complexity of Randomized SVD on $\widetilde{A}$ is $O(ep + np^2)$, that of constructing $E$ and $e$ is $O(ep)$, and that of the message passing framework in Equation (9) is $O(np^2)$. Thus, the overall time complexity of SDE-GNN is $O(ep + np^2)$, where $n$ and $e$ denote the number of nodes and edges respectively, and $p$ denotes the reduced dimension. Given that the number of layers $L$, the power of adjacency matrix $k$, the maximum order of polynomial expansion $m$, and the node feature dimension $d$ are small constants, their contributions are omitted from the complexity analysis. We compare SDE-GNN's time complexity with other distance-enhanced GNNs in Table 1.

## 4 THEORETICAL ANALYSIS

In this section, we compare the expressive power of the proposed method with existing DE-GNNs. Specifically, we adopt the *Generalized Distance Weisfeiler-Lehman* (GD-WL) test (Zhang et al., 2023) as a theoretical abstraction of various DE-GNNs and compare the corresponding GD-WL variants in terms of their ability to distinguish non-isomorphic graphs. For a given graph $\mathcal{G}(\mathcal{V}, \mathcal{E})$, GD-WL iteratively refines node colors according to the update rule

$$\chi_G^{t+1}(u) = \text{hash}\left(\{\{(\chi_G^t(v), d_G(u, v)) : v \in V\}\}\right),\tag{10}$$

where $\chi_G^t(u)$ denotes the color of node $u$ at round $t$, hash$(\cdot)$ is an injective function that assigns a distinct color to each unique input, and $d_G(\cdot, \cdot)$ is a distance feature mapping that maps a pair of nodes to a distance feature, which will be described in detail later. After running GD-WL for a sufficiently large round $T$, it determines whether two graphs $\mathcal{G}_1(\mathcal{V}_1, \mathcal{E}_1)$ and $\mathcal{G}_2(\mathcal{V}_2, \mathcal{E}_2)$ are isomorphic by comparing the multisets of node colors $\{\{\chi_{G_1}^T(u) : u \in \mathcal{V}_1\}\}$ and $\{\{\chi_{G_2}^T(u) : u \in \mathcal{V}_2\}\}$. Intuitively, Equation (10) can be viewed as a global GNN augmented with distance features, where each node $u$ updates its representation $\chi_G^{t+1}(u)$ by aggregating information from all nodes $v \in \mathcal{V}$ in the graph, with each message comprising both the representation $\chi_G^t(v)$ and the corresponding distance feature $d_G(u, v)$. By specifying different distance feature mapping $d$, Equation (10) can instantiate various DE-GNNs. For instance, for DE-GNNs that construct additional edge features $\boldsymbol{f}_{u,v} \in \mathbb{R}^D$ on existing edges, we can define the distance as $d_G(u, v) = [1, \boldsymbol{f}_{u,v}] \in \mathbb{R}^{D+1}$ for $\{u, v\} \in E$, and set $d_G(u, v)$ to a zero vector in $\mathbb{R}^{D+1}$ otherwise. More details about the GD-WL can be found in the Appendix B.

**Distance Feature Mapping.** Given a graph $\mathcal{G}(\mathcal{V}, \mathcal{E})$, the distance feature mapping $d$ outputs a function $d_G : V \times V \rightarrow \mathbb{R}^{D_G}$ that maps a pair of nodes within $V$ into a distance vector. The dimension of the distance vector $D_G$ can vary across graphs. (Zhang et al., 2024) shows that GD-WL equipped with the eigenspace projection distance achieves the highest expressiveness among commonly used distances such as resistance distance and shortest path distance. To illustrate the advantages of our method over existing DE-GNNs, we adopt GD-WL with the adjacency-power-based distance $d^{AP}$ as a proxy, and compare its expressiveness against that of GD-WL with the eigenspace projection distance $d^{EP}$. Specifically, given a graph $\mathcal{G}(\mathcal{V}, \mathcal{E})$, let $\tilde{\boldsymbol{L}} = \boldsymbol{D}^{-1/2}\boldsymbol{L}\boldsymbol{D}^{-1/2}$ denote the normalized Laplacian matrix and $\tilde{\boldsymbol{L}} = \sum_{i=1}^m \lambda_i \boldsymbol{P}_i$ denotes its spectral decomposition, where $\lambda_1, \cdots, \lambda_m$ are the eigenvalues of $\tilde{\boldsymbol{L}}$ and $\boldsymbol{P}_1, .., \boldsymbol{P}_m \in \mathbb{R}^{|V| \times |V|}$ are the corresponding eignspace projections. The eigenspace projection distance between two nodes $u, v \in V$ is

$$d_G^{EP}(u, v) = [\lambda_1, (P_1)_{u,v}, \lambda_2, (P_2)_{u,v}, \cdots, \lambda_m, (P_m)_{u,v}] \in \mathbb{R}^{2m},\tag{11}$$

where $(P_i)_{u,v}$ denotes the u-th row and v-th column of $\boldsymbol{P}_i$. Let $\tilde{\boldsymbol{A}} = \tilde{\boldsymbol{D}}^{-1/2}\boldsymbol{A}\tilde{\boldsymbol{D}}^{-1/2}$ be the normalized adjacency matrix of the given graph $\mathcal{G}$ and $n = |\mathcal{V}|$ be the number of nodes. The adjacency-powered-based distance feature between two nodes $u, v$ is defined as

$$d_G^{AP}(u, v) = [\tilde{A}_{u,v}^0, \tilde{A}_{u,v}^1, \cdots, \tilde{A}_{u,v}^n] \in \mathbb{R}^{n+1},\tag{12}$$

where $\tilde{\boldsymbol{A}}^i$ denotes the i-th power of $\tilde{\boldsymbol{A}}$.

In the following parts, we first establish the connection between GD-WL with $d^{AP}$ and $d^{EP}$ using Theorem 1, and then demonstrate that GD-WL with $d^{AP}$ bounds the expressiveness of GD-WL with $d^{EP}$ using Theorem 2. The Proofs of Theorem 1 and Theorem 2 can be found in Appendix C.1 and Appendix C.2, respectively.

**Theorem 1** *Let $\sigma(\boldsymbol{M}) = \{\{\lambda_1, \cdots, \lambda_n\}\}$ denote the multiset of eigenvalues of a matrix $\boldsymbol{M} \in \mathbb{R}^{n \times n}$, and let $\tilde{\boldsymbol{A}}_G = \boldsymbol{D}_G^{-1/2}\boldsymbol{A}_G\boldsymbol{D}_G^{-1/2}$ be the normalized adjacency matrix of a graph $\mathcal{G}$. For any two*

graphs $\mathcal{G}_1(\mathcal{V}_1, \mathcal{E}_1)$ and $\mathcal{G}_2(\mathcal{V}_2, \mathcal{E}_2)$, if they cannot be distinguished by GD-WL equipped with the adjacency-power-based distance $d^{AP}$, then the following must hold:

- $|\mathcal{V}_1| = |\mathcal{V}_2|$ and $\sigma(\tilde{\boldsymbol{A}}_{\mathcal{G}_1}) = \sigma(\tilde{\boldsymbol{A}}_{\mathcal{G}_2})$

- if $d^{AP}_{G_1}(u, v) = d^{AP}_{G_2}(x, y)$ then $d^{EP}_{G_1}(u, v) = d^{EP}_{G_2}(x, y)$

**Remark 1.** While the power of the adjacency matrix has been used as a distance feature in prior works, its role in message passing remains insufficiently understood. Unlike previous case-specific analyses (Ma et al., 2023; Li et al., 2020), Theorem 1 offers a principled understanding. Specifically, the first statement shows that if two graphs $\mathcal{G}_1$ and $\mathcal{G}_2$ cannot be distinguished by GD-WL with $d^{\mathrm{AP}}$, then their adjacency spectra must be identical, i.e., $\sigma(\tilde{\boldsymbol{A}}_{\mathcal{G}_1}) = \sigma(\tilde{\boldsymbol{A}}_{\mathcal{G}_2})$. Therefore, the features (i.e., the final multiset of node colors) extracted by GD-WL with $d^{\mathrm{AP}}$ encode the graph spectra, which partition graphs into different equivalence classes that share the same adjacency spectrum. The second statement serves as a bridge connecting $d^{\mathrm{AP}}$ to the eigenspace projection distance $d^{\mathrm{EP}}$, thereby laying the foundation for comparing the two GD-WL variants in Theorem 2.

**Theorem 2** *For any two graphs $\mathcal{G}_1$ and $\mathcal{G}_2$ that cannot be distinguished by GD-WL equipped with $d^{\mathrm{AP}}$, they also cannot be distinguished by GD-WL equipped with $d^{\mathrm{EP}}$.*

**Remark 2.** From Theorem 2, we know that if two graphs cannot be distinguished by GD-WL with $d^{\mathrm{AP}}$, then they also cannot be distinguished by GD-WL with $d^{\mathrm{EP}}$. Conversely, those distinguishable by $d^{\mathrm{EP}}$ are also distinguishable by $d^{\mathrm{AP}}$. This theorem plays a key role in characterizing the expressiveness of GD-WL with $d^{AP}$. It establishes that GD-WL with $d^{AP}$ upper-bounds the expressiveness of GD-WL with other distance metrics, as $d^{EP}$ is known to achieve the highest expressiveness among commonly used metrics. Consequently, GD-WL with $d^{AP}$ inherits known expressiveness results of GD-WL, such as being strictly more powerful than 1-WL (Zhang et al., 2024) and can distinguish block cut-vertex/edge trees (Zhang et al., 2023). This positions $d^{AP}$ within the broader landscape of expressiveness studies.

## 5 EXPERIMENTAL RESULTS

### 5.1 EXPERIMENT SETTING

We mainly perform experiments on 17 widely used graph datasets, including 14 node classification datasets and 3 graph classification datasets. We compare our model with 15 state-of-the-art baselines, including 4 classic graph neural networks, 4 efficient graph transformer models, and 7 expressive graph neural networks. Our code is released anonymously at `https://anonymous.4open.science/r/SDE-GNN-7853`. More information about datasets, baselines, and experimental settings like hyperparameter configurations can be found in Appendix D.

### 5.2 PERFORMANCE COMPARISON

**Effectiveness.** As shown in Table 2, 3, and 4, SDE-GNN model yields consistently strong performance across all 17 datasets with different properties, which achieve the best performance on 11 out of them, demonstrating the effectiveness of the proposed distance-enhanced message passing framework. Besides the above observations, specific strengths of SDE-GNN can also be concluded from the performance of different methods: 1) In Table 2, the rank of SDE-GNN is significantly lower than that of the second one, Exphormer, on the five datasets that exhibit complex structural patterns, empirically demonstrating its strong expressive power. 2) In Table 3, SDE-GNN achieves the best performance on two heterophilous datasets, indicating its ability to capture high-order structural patterns—a key factor in effectively modeling heterophilous graphs. 3) In Table 4, SDE-GNN achieves the highest average rank on three graph classification datasets, demonstrating its effectiveness in capturing the overall graph structure.

**Efficiency and Scalability.** For scalability, the results from Table 2 and Table 3 show that SDE-GNN successfully scales to large-scale graphs, with node counts ranging from over 100K to approximately 3 million (e.g., ogbn-arxiv, twitch-gamers, pokec, and snap-patents). While existing expressive GNNs can only run on small graphs with a few thousand nodes and will encounter the out-of-memory error

Table 2: Test performance on 5 complex structural datasets is presented as the mean ± standard deviation over 5 runs with different random seeds. "OOM" indicates out of memory. Highlighted are the top **first** and **second** results. "Acc" denotes the test accuracy metric, and "Rank" denotes the average rank across different datasets.

| Model | CLUSTER | PATTERN | USA-Airports | Europe-Airports | Brazil-Airports | Rank ↓ |
|---|---|---|---|---|---|---|
| | Acc ↑ | Acc ↑ | Acc ↑ | Acc ↑ | Acc ↑ | |
| GCN | 67.46 ± 0.84 | 84.37 ± 0.03 | 64.03 ± 0.96 | 60.34 ± 2.53 | 75.23 ± 2.14 | 7.80 |
| GraphSAGE | 64.28 ± 0.75 | 82.46 ± 0.12 | 63.46 ± 1.13 | 58.42 ± 3.16 | 77.77 ± 3.32 | 9.20 |
| GAT | 69.53 ± 0.53 | 78.68 ± 0.10 | 63.98 ± 2.06 | 59.67 ± 3.44 | 77.89 ± 3.45 | 7.60 |
| GIN | 65.31 ± 0.96 | 84.79 ± 0.03 | 64.04 ± 0.88 | 59.12 ± 1.84 | 75.56 ± 2.56 | 7.80 |
| GraphGPS | 78.03 ± 0.21 | 86.19 ± 0.06 | 43.10 ± 2.44 | 46.35 ± 0.52 | 52.22 ± 1.57 | 10.80 |
| Exphormer | 78.07 ± 0.04 | 86.74 ± 0.02 | 60.96 ± 1.22 | 56.20 ± 1.03 | 76.67 ± 1.57 | 6.60 |
| SGFormer | 59.76 ± 0.86 | 83.31 ± 0.15 | 61.53 ± 1.11 | 50.95 ± 1.40 | 69.78 ± 3.37 | 12.20 |
| GOAT | 58.61 ± 0.67 | 82.76 ± 0.11 | 38.65 ± 2.41 | 36.77 ± 3.88 | 37.78 ± 4.24 | 14.60 |
| DE-GNN | 76.56 ± 0.92 | 85.21 ± 0.08 | 64.16 ± 0.95 | 60.69 ± 2.73 | 76.47 ± 2.03 | 5.80 |
| Graphormer | 76.79 ± 1.13 | 85.58 ± 0.16 | 63.74 ± 1.03 | 58.72 ± 2.87 | 74.31 ± 2.31 | 7.60 |
| ESAN | OOM | OOM | OOM | OOM | OOM | 16.00 |
| GRIT | 79.43 ± 0.33 | 86.62 ± 0.08 | 63.36 ± 1.53 | 53.29 ± 2.76 | 71.11 ± 2.38 | 7.20 |
| RD-WL | 78.22 ± 0.98 | 86.35 ± 0.18 | 66.53 ± 1.24 | 61.48 ± 3.03 | 77.28 ± 3.75 | 3.20 |
| SPE | 70.37 ± 0.45 | 85.73 ± 0.05 | 56.28 ± 2.47 | 51.37 ± 3.53 | 68.79 ± 5.03 | 10.60 |
| NeuralWalker | 78.19 ± 0.20 | 86.98 ± 0.02 | 54.78 ± 2.62 | 56.20 ± 2.47 | 73.33 ± 1.26 | 7.40 |
| SDE-GNN (ours) | 78.68 ± 0.24 | 86.82 ± 0.03 | 67.57 ± 1.03 | 63.02 ± 0.84 | 82.22 ± 1.42 | 1.40 |

Table 3: Test performance on 5 real-world medium-scale datasets and 4 large-scale datasets is presented as the mean ± standard deviation over 5 runs with different random seeds. "OOM" indicates out of memory. Highlighted are the top **first** and **second** results. "Acc" denotes the test accuracy metric, and "Rank" denotes the average rank across datasets.

| Model | Cora | CiteSeer | PubMed | Squirrel | Chameleon | ogbn-arxiv | twitch-gamers | pokec | snap-patents | Rank ↓ |
|---|---|---|---|---|---|---|---|---|---|---|
| | Acc ↑ | Acc ↑ | Acc ↑ | Acc ↑ | Acc ↑ | Acc ↑ | Acc ↑ | Acc ↑ | Acc ↑ | |
| GCN | 85.10 ± 0.48 | 72.94 ± 0.41 | 80.96 ± 0.49 | 45.04 ± 1.59 | 45.55 ± 3.35 | 73.11 ± 0.23 | 63.39 ± 0.34 | 85.20 ± 0.23 | 46.17 ± 0.18 | 2.22 |
| GraphSAGE | 83.68 ± 0.50 | 72.04 ± 0.56 | 78.66 ± 0.50 | 40.56 ± 1.46 | 44.08 ± 4.62 | 72.98 ± 0.27 | 62.18 ± 0.42 | 84.78 ± 0.27 | 45.62 ± 0.34 | 6.33 |
| GAT | 82.70 ± 0.53 | 70.96 ± 0.68 | 80.42 ± 0.66 | 40.78 ± 1.95 | 41.27 ± 4.06 | 73.26 ± 0.25 | 60.23 ± 0.53 | 79.35 ± 0.65 | 43.76 ± 1.14 | 7.00 |
| GIN | 83.82 ± 0.62 | 71.43 ± 0.70 | 79.70 ± 0.59 | 43.24 ± 1.74 | 44.26 ± 4.28 | 71.68 ± 0.31 | 61.85 ± 0.47 | 78.56 ± 0.42 | 45.88 ± 0.71 | 6.11 |
| GraphGPS | 83.81 ± 0.93 | 72.64 ± 1.35 | 79.88 ± 0.37 | 39.82 ± 2.14 | 41.55 ± 4.06 | 71.22 ± 0.62 | 61.59 ± 0.58 | OOM | OOM | 8.00 |
| Exphormer | 83.24 ± 1.23 | 71.75 ± 1.32 | 79.62 ± 0.83 | 39.10 ± 1.02 | 44.88 ± 2.86 | 72.14 ± 0.48 | OOM | OOM | OOM | 8.33 |
| SGFormer | 84.50 ± 0.82 | 72.54 ± 0.23 | 80.31 ± 0.58 | 41.85 ± 2.41 | 44.93 ± 3.72 | 72.68 ± 0.16 | 65.92 ± 0.19 | 76.17 ± 1.86 | 40.13 ± 2.13 | 5.11 |
| GOAT | 76.83 ± 1.07 | 54.30 ± 1.25 | 78.64 ± 0.69 | 41.15 ± 1.86 | 42.27 ± 3.87 | 71.88 ± 0.42 | 63.38 ± 0.26 | 71.43 ± 3.47 | 42.53 ± 3.65 | 8.89 |
| DE-GNN | 84.67 ± 0.53 | 71.69 ± 0.43 | OOM | 44.42 ± 1.93 | 46.51 ± 3.64 | OOM | OOM | OOM | OOM | 7.44 |
| Graphormer | 82.43 ± 0.86 | 70.16 ± 0.77 | OOM | 41.87 ± 1.86 | 43.77 ± 4.13 | OOM | OOM | OOM | OOM | 9.44 |
| ESAN | OOM | OOM | OOM | OOM | OOM | OOM | OOM | OOM | OOM | 11.89 |
| GRIT | 79.45 ± 0.78 | 68.77 ± 0.89 | OOM | 39.57 ± 1.58 | 44.25 ± 3.69 | OOM | OOM | OOM | OOM | 10.56 |
| RD-WL | 84.76 ± 0.94 | 72.86 ± 0.92 | OOM | 44.69 ± 2.05 | 47.03 ± 3.82 | OOM | OOM | OOM | OOM | 6.56 |
| SPE | OOM | OOM | OOM | OOM | OOM | OOM | OOM | OOM | OOM | 11.78 |
| NeuralWalker | 82.10 ± 1.23 | 69.52 ± 1.06 | 76.70 ± 0.83 | 40.71 ± 2.33 | 43.30 ± 4.59 | 64.59 ± 2.64 | 53.33 ± 0.94 | OOM | OOM | 9.89 |
| SDE-GNN (ours) | 85.92 ± 0.42 | 73.83 ± 0.55 | 80.63 ± 0.51 | 46.32 ± 1.53 | 47.24 ± 3.56 | 72.90 ± 0.26 | 66.04 ± 0.35 | 85.23 ± 0.32 | 46.80 ± 0.25 | 1.44 |

on these larger graphs, highlighting the superior scalability of SDE-GNN over existing expressive GNNs. To further validate the efficiency advantage of SDE-GNN over existing expressive and efficient GNNs, we report the running time, memory usage, and accuracy of different methods on twitch-gamers dataset in Figure 2. We exclude NeuralWalker as its performance is not competitive on this dataset. As shown in Figure 2, SDE-GNN achieves the highest accuracy and the lowest running time, while maintaining a moderate level of memory consumption, demonstrating its overall efficiency.

## 5.3 ABLATION STUDIES

To verify the effectiveness of the proposed components, we compare SDE-GNN with the following two variants of 1) w/o polynomial expansion by keeping only the first order in Equation (3) and 2) w/o separation that also computes the first order term in Equation (9) using dimension reduction. As shown in Table 5, there is a significant performance drop of w/o separation, highlighting the advantage of performing computations explicitly rather than relying on dimensionality reduction. The performance of w/o separation is N/A on CLUSTER and PATTERN because the maximum number of nodes in these datasets does not exceed 200, and dimension reduction is not applied. For

Table 4: Test performance on 3 real-world graph-classification datasets.

| Model | IMDB-BINARY | IMDB-MULTI | REDDIT-BINARY | Rank ↓ |
|---|---|---|---|---|
| | Accuracy ↑ | Accuracy ↑ | Accuracy ↑ | |
| GCN | 70.66 ±0.95 | 46.77 ±1.41 | 75.49 ±1.11 | 9.67 |
| GraphSAGE | 69.92 ±1.24 | 46.14 ±1.44 | 75.03 ±1.23 | 11.33 |
| GAT | 70.87 ±1.38 | 47.23 ±1.59 | 74.87 ±1.58 | 10.00 |
| GIN | 70.95 ±1.15 | 50.32 ±1.53 | 75.16 ±1.52 | 6.67 |
| GraphGPS | 66.36 ±1.43 | 41.77 ±1.67 | 86.38 ±1.64 | 11.67 |
| Exphormer | 65.48 ±1.51 | 41.96 ±1.88 | 87.04 ±1.75 | 11.33 |
| SGFormer | 64.25 ±1.08 | 42.23 ±1.45 | 85.73 ±1.42 | 12.33 |
| GOAT | 65.36 ±2.13 | 41.75 ±2.38 | 83.53 ±2.18 | 13.33 |
| DE-GNN | 70.93 ±1.02 | 48.49 ±1.85 | **91.97** ±1.17 | 4.67 |
| Graphormer | 67.67 ±1.38 | 45.69 ±2.34 | 87.79 ±2.32 | 9.67 |
| ESAN | 73.05 ±1.84 | **50.78** ±1.95 | OOM | 6.00 |
| GRIT | 72.13 ±1.37 | 49.56 ±1.74 | 89.64 ±2.31 | 3.67 |
| RD-WL | 69.48 ±2.16 | 49.86 ±1.64 | 89.45 ±1.63 | 6.33 |
| SPE | 70.44 ±1.67 | 48.12 ±1.83 | OOM | 10.67 |
| NeuralWalker | 72.00 ±1.16 | 48.64 ±1.77 | 79.66 ±1.45 | 6.67 |
| SDE-GNN (ours) | **73.33** ±1.34 | 50.66 ±1.62 | 91.78 ±1.48 | **1.67** |

Table 5: Ablation study results.

| Model | CLUSTER | PATTERN | ogbn-arxiv | twitch-gamers |
|---|---|---|---|---|
| | Accuracy ↑ | Accuracy ↑ | Accuracy ↑ | Accuracy ↑ |
| SDE-GNN | 78.68 | 86.82 | 72.90 | 66.04 |
| w/o polynomial | 78.12 | 86.74 | 72.81 | 65.91 |
| w/o separation | N/A | N/A | 69.91 | 64.83 |

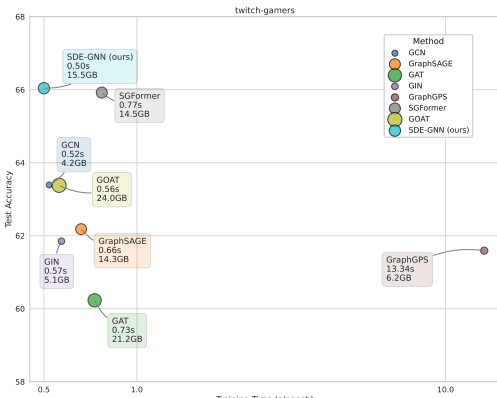

Figure 2: Comparison of training time (s/epoch) and test accuracy (%) of methods on twitch-gamers dataset (with 168,114 nodes). The size of the node reflects the GPU memory cost.

Table 6: The trade-offs between test accuracy (%), runtime (s/epoch), and GPU memory usage (MB) by varying the projection dimension $p$ on twitch-gamers dataset (with 168,114 nodes). "rank" indicates SDE-GNN's test accuracy ranking compared to other baseline methods.

| Metric \Param $p$ | 100 | 200 | 300 | 400 | 500 | 600 | 700 | 800 | 900 | 1000 |
|---|---|---|---|---|---|---|---|---|---|---|
| Runtime (s/epoch) | 0.38 | 0.40 | 0.42 | 0.44 | 0.46 | 0.48 | 0.50 | 0.52 | 0.54 | 0.54 |
| GPU Memory (MB) | 6,204 | 7,094 | 7,874 | 8,764 | 9,788 | 10,320 | 11,344 | 11,708 | 13,732 | 15,460 |
| Test Acc | 65.03 | 65.42 | 65.74 | 65.82 | 65.88 | 65.96 | 66.02 | 65.94 | 66.02 | 66.04 |
| Rank | 2 | 2 | 2 | 2 | 2 | 1 | 1 | 1 | 1 | 1 |

w/o polynomial, its performance consistently drops on these four datasets, verifying the necessity of adopting the polynomial term in the learning process of the distance feature mapping. Besides, as people focus on the trade-off between effectiveness and cost, we resort to empirical analysis to gain insights into how dimensionality (i.e., $p$) affects performance. In Table 6, we vary from $p = 100$ to $1,000$ on twitch-gamers dataset and report the accuracy, runtime, GPU memory usage, and the rank of our method among all compared baselines. From the results, we conclude that both performance and the required computational resources generally increase as dimensionality grows, but at different rates: $p$ has a much smaller impact on performance compared to memory usage and runtime. For example, as $p$ decreases from 1,000 to 100, memory and runtime decrease by 60% and 30%, respectively. While performance drops by only 1.01, and the rank drops by just one. For practical usage, a dimensionality between 100 and 300 may provide a favorable balance between performance and efficiency. More experimental results of the time and memory consumption at different node scales and the analysis of dimensionality reduction methods are deferred to Appendix E.

## 6 CONCLUSION

In this work, we have addressed the critical challenge of enhancing the expressiveness of DE-GNNs while ensuring scalability to large graphs. By proposing SDE-GNN, a scalable distance-enhanced GNN framework leveraging adjacency-power-based distance features, learnable polynomial distance encoding, and randomized dimensionality reduction, we significantly reduce computational complexity from quadratic to near-linear with respect to graph size. Our theoretical analysis establishes that the chosen distance features possess superior expressive power, exceeding that of commonly used alternatives. Extensive empirical evaluations across diverse benchmark datasets validate that SDE-GNN consistently outperforms strong baselines in accuracy, efficiency, and scalability. This study not only advances the understanding of distance features in GNNs but also offers a practical solution for applying expressive GNN models to large-scale graph mining tasks.

## 7 ETHICS STATEMENT

The datasets used for the experiments were publicly available and fully anonymized. We rigorously evaluated our model for potential biases, societal implications, and the safety of its generated content. The authors declare no conflicts of interest. For transparency and reproducibility, the code and data are open-sourced.

## 8 REPRODUCIBILITY STATEMENT

All experimental code and data are open-sourced available in an anonymous repository, allowing for the complete reproduction of our experimental results.

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

## A  EXAMPLES TO SHOW THE EXPRESSIVE ADVANTAGE OF DE-GNNs

The figure below demonstrates the expressive advantage of DE-GNNs through the task of triangle detection. In Figure 3 (b) and (c), vanilla GNNs fail to determine whether a node is part of a triangle because all nodes share the same local structure—each connected to two neighbors. As a result, standard message passing assigns identical representations to all 12 nodes, making it impossible to distinguish the blue and green nodes (which are part of triangles) from the orange ones (which are not). In contrast, the proposed distance-enhanced method determines whether a node is part of a triangle by checking if there exists another node that is both one-hop and two-hop away—an indicator of the triangle and is a relation captured by the distance features. Figure 3 (a) further illustrates the practical utility of triangle detection in a real-world citation network, where nodes represent papers. In this scenario, not all citations carry equal importance—some are core citations, while others are superficial. For example, the red nodes are core citations of the yellow node, as all papers citing the yellow one also cite the red ones. In contrast, the green nodes are superficial citations lacking this pattern. Triangle-aware expressiveness provides an effective means to capture such relationships, as it allows one to infer whether two cited nodes form a triangle and thus identify core versus superficial citations more effectively.

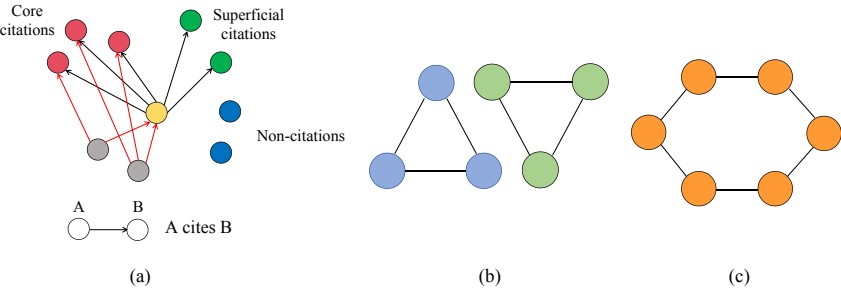

(a)  (b)  (c)

Figure 3: Examples to show the expressiveness of DE-GNNs. (a) depicts a real-world citation networks with core citations, superficial citations, and non-citations. (b) and (c) illustrate two regular graphs, where the objective is to detect nodes within triangles.

## B  INTRODUCTION OF GD-WL

**Notations**. We use $G(V, E)$ to denote a graph. We use $\mathcal{C}$ to denote a set of colors and hash$(\cdot)$ to note a bijective hash function that maps an arbitrary object $x$ into a color hash$(x) \in \mathcal{C}$. We use the notation $\{\{\}\}$ to denote a multiset. We use $d$ to denote a distance metric, where given a graph $G(V, E)$, it outputs a function $d_G : V \times V \to \mathbb{R}^{D_G}$ that maps a pair of nodes within $V$ into a distance vector. Here, the dimension of the distance vector $D_G$ may vary across graphs, and we will introduce it later in this section. A partition of a set $S$ is a collection $Q$ of nonempty, pairwise disjoint subsets of $S$ such that $\bigcup_{S' \in Q} S' = S$. We use $Q_1 \preceq Q_2$ to denote that partition $Q_1$ is finer than $Q_2$, where for every $S_1 \in Q_1$, there exists $S_2 \in Q_2$ such that $S_1 \subseteq S_2$. We denote $Q_1 \prec Q_2$ if $Q_1 \preceq Q_2$ and $Q_1 \neq Q_2$.

**GD-WL**. GD-WL is an extension of the WL-test for the graph isomorphism test, which incorporates a distance metric in the color refinement procedure. Specifically, given a graph $G(V, E)$, GD-WL begins by assigning a predefined $c \in \mathcal{C}$ to all nodes and iteratively applies the following formula to update the color of nodes

$$\chi_G^{t+1}(u) = \text{hash}(\{\{(\chi_G^t(v), d_G(u, v)) : v \in V\}\}), \tag{13}$$

where $\chi_G^t(u)$ denotes the color of node $u$ at round $t$. The set of node colors obtained at iteration $t$, denoted as $\{\{\chi_G^t(u) : u \in V\}\}$, can induce a partition of the node set as $Q^t(V) = \{(\chi_G^t)^{-1}(c) : c \in \mathcal{C}\}$, where $(\chi_G^t)^{-1}(c) = \{u \in V : \chi^t(u) = c\}$ represents the set of nodes assigned the same color $c$. The GD-WL will terminate at round $t + 1$, once the partition is stable, i.e., $Q^{t+1}(V) = Q^t(V)$. One important property of the partition is that the obtained partition $\{Q^t(V)\}_{t \in \mathbb{N}}$ is a sequence of refinements, where there exists a $T \in \mathbb{N}$ such that $Q^t(V) \prec Q^{t+1}(V)$ for all $t \leq T$ and

$Q^t(V) = Q^{t+1}(V)$ for all $t \geq T^1$ Since the finest partition is $\{\{u\} : u \in V\}$, the $T$ will be less than $|V|$ and the GD-WL will terminate at finite step. To determine whether two graphs $G_1(V_1, E_1)$ and $G_2(V_2, E_2)$ are isomorphic, we need to run GD-WL on two graphs in parallel and compare their final node colors. In this situation, we use notation $Q^t(V_1 \cup V_2)$ to denote the partition of $V_1 \cup V_2$ at round $t$, where each element of $Q^t(V_1 \cup V_2)$ is a subset of $V_1 \cup V_2$ consisting of nodes that share the same color. Algorithm 2 outlines the detailed procedure by which GD-WL determines whether two graphs are isomorphic.

**Distance Feature Mapping.** Various choices for $d$ are possible. For instance, it can be the shortest-path distance, where $D_G \equiv 1$ and $d_G(u, v) \in \mathbb{R}^1$ is the shortest path from $u$ to $v$; or it can be a fixed-length random walk distance, where $D_G \equiv K$ and $d_G(u, v) = [p_{u,v}^1, \ldots, p_{u,v}^k] \in \mathbb{R}^k$, with $p_{u,v}^k$ denoting the probability of reaching $v$ from $u$ via a $k$-step random walk. (Zhang et al., 2024) demonstrates that GD-WL equipped with the eigenspace projection distance exhibits the highest expressiveness among commonly used distance feature mappings. Therefore, in our analysis, we focus on the eigenspace projection distance and our adjacency-power-based distance, denoting them by $d^{EP}$ and $d^{AP}$, respectively. Specifically, given a graph $G(V, E)$, let $\tilde{L} = D^{-1/2} L D^{-1/2}$ denote the normalized Laplacian matrix and $\tilde{L} = \sum_{i=1}^m \lambda_i P_i$ denotes its spectral decomposition, where $\lambda_1, \cdots, \lambda_m$ are the eigenvalues of $\tilde{L}$ and $P_1, .., P_m \in \mathbb{R}^{|V| \times |V|}$ are the corresponding eignspace projections. The eigenspace projection distance between two nodes $u, v \in V$ is

$$d_G^{EP}(u, v) = [\lambda_1, (P_1)_{u,v}, \lambda_2, (P_2)_{u,v}, \cdots, \lambda_m, (P_m)_{u,v}] \in \mathbb{R}^{2m}, \tag{14}$$

where $(P_i)_{u,v}$ denote the u-th row and v-th column of $P_i$. Let $\tilde{A} = \tilde{D}^{-1/2} A \tilde{D}^{-1/2}$ be the normalized adjacency matrix of the given graph $G$ and $n = |V|$ be the number of nodes. The adjacency-power-based distance feature between two nodes $u, v$ is defined as

$$d_G^{AP}(u, v) = [\tilde{A}_{u,v}^0, \tilde{A}_{u,v}^1, \cdots, \tilde{A}_{u,v}^n] \in \mathbb{R}^{n+1}, \tag{15}$$

where $\tilde{A}^i$ denotes the i-th power of $\tilde{A}$.

---

**Algorithm 2** The Generalized Distance Weisfeiler-Lehman Algorithm

---

**Require:** Graphs $G_1 = (V_1, E_1)$, graph $G_2 = (V_2, E_2)$, distance metric $d$
**Ensure:** Whether $G_1$ and $G_2$ are isomorphism
1: Initialize $t = 0$, $\chi_{G_1}^0(u) = c$ and $\chi_{G_2}^0(x) = c$ for all $u \in V_1$ and $x \in V_2$ correspondingly
2: **while** $True$ **do**
3:    $t \leftarrow t + 1$
4:    **for** $u \in V_1$ **do**
5:       $\chi_{G_1}^t(u) = \text{hash}(\{\{(d_{G_1}(u, v), \chi_{G_1}^{t-1}(v)) : v \in V_1\}\})$
6:    **end for**
7:    **for** $x \in V_2$ **do**
8:       $\chi_{G_2}^t(x) = \text{hash}(\{\{(d_{G_2}(x, y), \chi_{G_2}^{t-1}(y)) : y \in V_2\}\})$
9:    **end for**
10:   **if** $Q^t(V_1 \cup V_2) = Q^{t-1}(V_1 \cup V_2)$ **then**
11:      **break**
12:   **end if**
13: **end while**
14: **Return:** $\{\{\chi_{G_1}^t(u) : u \in V_1\}\} = \{\{\chi_{G_2}^t(x) : x \in V_2\}\}$

---

## C    Proofs of Theorems

In this section, we first give lemmas that are required for the proofs of Theorem 1 and Theorem 2, and then give the detailed proofs.

**Lemma 1** *For any $\lambda_1, \ldots \lambda_n \in \mathbb{C}$, any $\sigma_1, \ldots \sigma_n \in \mathbb{C}$, if $\sum_{i=1}^n \lambda_i^k = \sum_{i=1}^n \sigma_i^k$ holds for $0 \leq k \leq n$, then we will have $\{\{\lambda_1, \ldots \lambda_n\}\} = \{\{\sigma_1, \cdots, \sigma_n\}\}$.*

---

[1]More details can be found in Appendix B.1 of (Zhang et al., 2023).

*Proof.* Let $e_i(a_1, \cdots, a_n)$ denote $\sum_{1 \le j_1 < j_2 < \cdots < j_i \le n} (-1)^i a_{j_1} a_{j_2} \cdots a_{j_n}$, and $p_i(a_1, ..., a_n)$ denote $\sum_{j=1}^n a_j^i$. According to Newton's Identities, for $1 \le i \le n$, we have

$$e_i(a_1, ..., a_n) = \frac{-1}{i} \sum_{j=0}^{i-1} e_j(a_1, .., a_n) * p_{i-j}(a_1, .., a_n), \tag{16}$$

where $e_0(a_1, ..., a_n) = 0$. Thus if $p_i(\lambda_1, ..., \lambda_n) = p_i(\sigma_1, \sigma_2, ..., \sigma_n)$ for $0 \le i \le n$, then by recursion, we will have $e_i(\lambda_1, \cdots, \lambda_n) = e_i(\sigma_1, \cdots, \sigma_n)$ for $0 \le i \le n$. Then we can construct two polynomials $f(x) = \prod_{i=1}^n (x - \lambda_i)$, $g(x) = \prod_{i=1}^n (x - \sigma_i)$, where the coefficents of $x^{n-i}$ in $f(x)$ and $g(x)$ will be $e_i(\lambda_1, \cdots, \lambda_n)$ and $e_i(\sigma_1, \cdots, \sigma_n)$ respectively. Since $e_i(\lambda_1, \cdots, \lambda_n) = e_i(\sigma_1, \cdots, \sigma_n)$ for all $0 \le i \le n$, we will have $f(x) = g(x)$, so $\{\{\lambda_1, \cdots, \lambda_n\}\} = \{\{\sigma_1, \cdots, \sigma_n\}\}$.

**Lemma 2** *Let $\sigma(M)$ denote the multiset of eigenvalues for a matrix $M \in \mathbb{C}^{n \times n}$, and $tr(M)$ denote the trace of $M$. For any two matrices $X, Y \in \mathbb{R}^{n \times n}$ with $tr(X^i) = tr(Y^i)$ for $0 \le i \le n$, we will have $\sigma(X) = \sigma(Y)$.*

*Proof.* Let $x_1, \cdots, x_n$ denote the eigenvalues of $X$ and $y_1, \cdots, y_n$ denote the eigenvalues of $Y$. Since $tr(X^i) = tr(Y^i)$ for $0 \le i \le n$, we will have $\sum_{j=1}^n x_j^i = \sum_{j=1}^n y_j^i$ for $1 \le i \le n$. According to Lemma 1, we will have $\{\{x_1, \cdots, x_n\}\} = \{\{y_1, \cdots, y_n\}\}$.

### C.1 PROOF OF THEOREM 1

We refer Algorithm 2 as the specific execution procedure of GD-WL. Assume that Algorithm 2 runs for $T$ rounds. Since it cannot distinguish between $G_1$ and $G_2$, we will have $\{\{\chi_{G_1}^T(u) : u \in V_1\}\} = \{\{\chi_{G_2}^T(x) : x \in V_2\}\}$. Thus, the number of nodes must be the same, i.e., $|V_1| = |V_2|$. Furthermore, there exists a bijective mapping: $f : V_1 \to V_2$ that satisfies $\chi_{G_1}^T(u) = \chi_{G_2}^T(f(u))$. Considering any two nodes that $x = f(u)$, since $\chi_{G_1}^T(u) = \chi_{G_2}^T(x)$, we will have

$$\text{hash}(\{\{(\chi_{G_1}^{T-1}(v), d_{G_1}^{AP}(u, v)) : v \in V_1\}\}) = \text{hash}(\{\{(\chi_{G_2}^{T-1}(y), d_{G_2}^{AP}(x, y)) : y \in V_2\}\})$$
$$\overset{I_1}{\Longrightarrow} \{\{(\chi_{G_1}^{T-1}(v), d_{G_1}^{AP}(u, v)) : v \in V_1\}\} = \{\{(\chi_{G_2}^{T-1}(y), d_{G_2}^{AP}(x, y)) : y \in V_2\}\}. \tag{17}$$

$I_1$ comes from the injectivity of the hash function. Therefore, there exists a bijecive mapping $f_1 : V_1 \to V_2$ satisfy that $d_{G_1}^{AP}(u, v) = d_{G_2}^{AP}(x, f_1(v))$. Among all nodes $v \in \mathcal{V}_1$, the adjacency-power-based distance $d_{G_1}^{AP}(u, u)$ is the only one whose first entry equals one, i.e., $d_{G_1}^{AP}(u, u)[1] = 1$, since only the diagonal elements of $\tilde{A}_{G_1}^0 = I$ are nonzero. The same reasoning applies to $d_{G_2}^{AP}(x, y)$ for $y \in V_2$. Consequently, we have $d_{G_1}^{AP}(u, u) = d_{G_2}^{AP}(x, x)$. Based on the above reasoning, we have

$$\chi_{G_1}^T(u) = \chi_{G_2}^T(x) \Longrightarrow d_{G_1}^{AP}(u, u) = d_{G_2}^{AP}(x, x) \tag{18}$$

Then we will have

$$\{\{\chi_{G_1}^T(u) : u \in V_1\}\} = \{\{\chi_{G_2}^T(v) : v \in V_2\}\}$$
$$\overset{I_1}{\Longrightarrow} \sum_{u \in V_1} d_{G_1}^{AP}(u, u) = \sum_{x \in V_2} d_{G_2}^{AP}(x, x)$$
$$\overset{I_2}{\Longrightarrow} tr(\tilde{A}_{G_1}^i) = tr(\tilde{A}_{G_2}^i), 0 \le i \le n \tag{19}$$
$$\overset{I_3}{\Longrightarrow} \sigma(\tilde{A}_{G_1}) = \sigma(\tilde{A}_{G_2})$$

where $n = |V_1| = |V_2|$ is the number of nodes, $I_1$ comes by applying equation 18, $I_2$ comes from the observation that for a graph $G(V, E)$ with $|V| = m$, we have

$$\sum_{u \in V} d_G^{AP}(u, u)[i] = \sum_{j=1}^m \tilde{A}_G^i[j, j] = tr(\tilde{A}_G^i), 1 \le i \le m, \tag{20}$$

where $d_G^{AP}(u, u)[i]$ denotes the i-th element of $d_G^{AP}(u, u)$ and $\tilde{A}_G^i[j, j]$ denotes element in the $j$-th row and $j$-th column of $\tilde{A}_G$. $I_3$ comes by applying Lemma 2. This completes the proof of the first statement of Theorem 1.

Consider the second statement. According to the first statement, we know that $\tilde{A}_{G_1}$ and $\tilde{A}_{G_2}$ have the same eigenvalues. Let $m$ be the number of distinct eigenvalues, and denote them by $\lambda_1, \ldots, \lambda_m$. Since for any graph $G$ it holds that $\tilde{L}_G = I - \tilde{A}_G$, the values $1 - \lambda_1, \ldots, 1 - \lambda_m$ are also the distinct eigenvalues of both $\tilde{L}_{G_1}$ and $\tilde{L}_{G_2}$. We denote the corresponding eigenspace projections of $\tilde{L}_{G_1}$ by $P_{G_1,1}, \ldots, P_{G_1,m}$, and those of $\tilde{L}_{G_2}$ by $P_{G_2,1}, \ldots, P_{G_2,m}$. Then, since $d_{G_1}^{AP}(u, v) = d_{G_2}^{AP}(x, y)$, we have

$$
\begin{aligned}
& d_{G_1}^{AP}(u, v) = d_{G_2}^{AP}(x, y) \Longrightarrow \tilde{A}_{G_1}^i[u, v] = \tilde{A}_{G_2}^i[u, v], \quad 0 \le i \le n \\
& \Longrightarrow \tilde{L}_{G_1}^i[u, v] = \tilde{L}_{G_2}^i[u, v], \quad 0 \le i \le n \\
& \overset{I_1}{\Longrightarrow} \sum_{j=1}^m \lambda_j^i P_{G_1,j}[u, v] = \sum_{j=1}^m \lambda_j^i P_{G_2,j}[u, v], \quad 0 \le i \le n \\
& \Longrightarrow \underbrace{\begin{bmatrix} 1 & 1 & \cdots & 1 \\ \lambda_1 & \lambda_2 & \cdots & \lambda_m \\ \lambda_1^2 & \lambda_2^2 & \cdots & \lambda_m^2 \\ \vdots & \vdots & \ddots & \vdots \\ \lambda_1^n & \lambda_2^n & \cdots & \lambda_m^n \end{bmatrix}}_{V} \cdot \begin{bmatrix} P_{G_1,1}[u, v] - P_{G_2,1}[u, v] \\ P_{G_1,2}[u, v] - P_{G_2,2}[u, v] \\ \vdots \\ P_{G_1,m}[u, v] - P_{G_2,m}[u, v] \end{bmatrix} = \mathbf{0} \quad (21) \\
& \overset{I_2}{\Longrightarrow} \begin{bmatrix} P_{G_1,1}[u, v] \\ P_{G_1,2}[u, v]] \\ \vdots \\ P_{G_1,m}[u, v] \end{bmatrix} = \begin{bmatrix} P_{G_2,1}[u, v] \\ P_{G_2,2}[u, v]] \\ \vdots \\ P_{G_2,m}[u, v] \end{bmatrix} \\
& \overset{I_3}{\Longrightarrow} d_{G_1}^{EP}(u, v) = d_{G_2}^{EP}(x, y),
\end{aligned}
$$

where $I_1$ follows from the properties of spectral decomposition, and $I_2$ holds because the columns of $V$ are linearly independent (the submatrix formed by the first $m$ rows of $V$ is the transpose of a Vandermonde matrix) and $I_3$ comes from the definition of $d^{EP}$.

## C.2 PROOF OF THEOREM 2

For notational simplicity, let $\chi_G^t(u)$ and $\overline{\chi}_G^t(u)$ denote the color of node $u$ in graph $G$ at round $t$ of Algorithm 2 using $d^{AP}$ and $d^{EP}$, respectively. Given two graphs $G_1(V_1, E_1)$ and $G_2(V_2, E_2)$, let $t_1$ denote the final round of Algorithm 2 with $d^{AP}$. We first show that for any two nodes $u \in V_1$ and $x \in V_2$, if $\chi_{G_1}^{t_1}(u) = \chi_{G_2}^{t_1}(x)$, we will have $\overline{\chi}_{G_1}^t(u) = \overline{\chi}_{G_2}^t(x)$ for all $t \in \mathbb{N}$, which is

$$
\chi_{G_1}^{t_1}(u) = \chi_{G_2}^{t_1}(x) \Longrightarrow \overline{\chi}_{G_1}^t(u) = \overline{\chi}_{G_2}^t(x), t \in \mathbb{N} \quad (22)
$$

We prove this by mathematical induction. At $t = 0$, all nodes have the same color, and Equation (22) holds. Assume Equation (22) holds at round $t$. Since $t_1$ is the final round, the node partition is stable after round $t_1$, and we will have

$$
\begin{aligned}
& \chi_{G_1}^{t_1}(u) = \chi_{G_2}^{t_1}(x) \Longrightarrow \chi_{G_1}^{t_1+1}(u) = \chi_{G_2}^{t_1+1}(x) \\
& \overset{I_1}{\Longrightarrow} \{\!\{(\chi_{G_1}^{t_1}(v), d_{G_1}^{AP}(u, v)) : v \in V_1\}\!\} = \{\!\{(\chi_{G_2}^{t_1}(y), d_{G_2}^{AP}(x, y)) : y \in V_2\}\!\} \\
& \overset{I_2}{\Longrightarrow} \{\!\{(\overline{\chi}_{G_1}^t(v), d_{G_1}^{EP}(u, v)) : v \in V_1\}\!\} = \{\!\{(\overline{\chi}_{G_2}^t(y), d_{G_2}^{EP}(x, y)) : y \in V_2\}\!\} \\
& \Longrightarrow \overline{\chi}_{G_1}^{t+1}(u) = \overline{\chi}_{G_2}^{t+1}(x),
\end{aligned} \quad (23)
$$

where $I_1$ comes from the definiton of $\chi_{G_1}^{t_1+1}(u)$ and $\chi_{G_2}^{t_1+1}(x)$, $I_2$ comes from the condition that Equation (22) holds at round $t$ and the second statement of Theorem 1. This finishes the proof of Equation (22). Let $t_2$ denote the final round of Algorithm 2 with $d^{EP}$. If Algorithm 2 with $d^{AP}$ can not distinguish $G_1$ and $G_2$, by applying Equation (22), we will have

$$
\begin{aligned}
& \{\!\{\chi_{G_1}^{t_1}(u) : u \in V_1\}\!\} = \{\!\{\chi_{G_2}^{t_1}(x) : x \in V_2\}\!\} \\
& \Longrightarrow \{\!\{\overline{\chi}_{G_1}^{t_2}(u) : u \in V_1\}\!\} = \{\!\{\overline{\chi}_{G_2}^{t_2}(x) : x \in V_2\}\!\},
\end{aligned} \quad (24)
$$

which finishes the proof of Theorem 2.

# D  EXPERIMENTAL DETAILS

## D.1  DATASETS

Table 7: Overview of the graph datasets used in this study.

| Dataset | # Graphs | Avg. # nodes | Avg. # edges | # Classes | Prediction task | Metric |
|---|---|---|---|---|---|---|
| PATTERN | 14,000 | 118.9 | 3,039.3 | 2 | Node Classification | Accuracy |
| CLUSTER | 12,000 | 117.2 | 2,150.9 | 6 | Node Classification | Accuracy |
| USA-Airports | 1 | 1,190 | 13,599 | 4 | Node Classification | Accuracy |
| Europe-Airports | 1 | 399 | 5,995 | 4 | Node Classification | Accuracy |
| Brazil-Airports | 1 | 131 | 1,074 | 4 | Node Classification | Accuracy |
| Cora | 1 | 2,708 | 10,556 | 7 | Node Classification | Accuracy |
| CiteSeer | 1 | 3,327 | 9,104 | 6 | Node Classification | Accuracy |
| PubMed | 1 | 19,717 | 88,648 | 3 | Node Classification | Accuracy |
| Squirrel | 1 | 2,223 | 46,998 | 5 | Node Classification | Accuracy |
| Chameleon | 1 | 2,277 | 36,101 | 5 | Node Classification | Accuracy |
| ogbn-arxiv | 1 | 169,343 | 1,166,243 | 40 | Node Classification | Accuracy |
| twitch-gamers | 1 | 168,114 | 6,797,557 | 2 | Node Classification | Accuracy |
| pokec | 1 | 1,632,803 | 30,622,564 | 2 | Node Classification | Accuracy |
| snap-patents | 1 | 2,923,922 | 13,975,788 | 5 | Node Classification | Accuracy |
| IMDB-BINARY | 1,000 | 19.8 | 193.1 | 2 | Graph Classification | Accuracy |
| IMDB-MULTI | 1,500 | 13.0 | 131.8 | 3 | Graph Classification | Accuracy |
| REDDIT-BINARY | 2,000 | 429.6 | 995.5 | 2 | Graph Classification | Accuracy |

**CLUSTER and PATTERN** (Dwivedi et al., 2023) are synthetic datasets sampled from Stochastic Block Model. Unlike other datasets, the prediction task here is an inductive node-level classification. In PATTERN the task is to recognize which nodes in a graph belong to one of 100 possible sub-graph patterns that were randomly generated with different SBM parameters than the rest of the graph. In CLUSTER, every graph is composed of 6 SBM-generated clusters, each drawn from the same distribution, with only a single node per cluster containing a unique cluster ID. The task is to infer which cluster ID each node belongs to.

**USA-Airports, Europe-Airports, and Brazil-Airports** (Ackland et al., 2005) are three air traffic networks, which were collected from the government websites throughout the year 2016 and were used to evaluate algorithms to learn structural representations of nodes. Networks are built such that nodes represent airports and there exists an edge between two nodes if there are commercial flights between them. In each dataset, the airports are divided into 4 different levels according to the annual passengers flow distribution by 3 quantiles: 25%, 50%, 75%. The goal is to infer the level of an airport using solely the connectivity pattern of them.

**Cora, CiteSeer, and PubMed** (Yang et al., 2016) are three text classification datasets. Each dataset contains bag-of-words representation of documents and citation links between the documents. The bag-of-words are embedded as feature vectors $x$, and the graph is constructed based on the citation links. The goal is to classify each document into one class.

**Squirrel and Chameleon** (Rozemberczki et al., 2021) are two heterophilous graph datasets. They are page-page networks on specific topics in Wikipedia. In those datasets, nodes represent web pages and edges are mutual links between pages. And node features correspond to several informative nouns in the Wikipedia pages. We classify the nodes into five categories in term of the number of the average monthly traffic of the web page. In this study, we adopt their filtered version described in (Luo et al., 2024).

**ogbn-arxiv** (Hu et al., 2020) is a directed graph, representing the citation network between all Computer Science (CS) arXiv papers indexed by MAG. Each node is an arXiv paper and each directed edge indicates that one paper cites another one. Each paper comes with a 128-dimensional feature vector obtained by averaging the embeddings of words in its title and abstract. The embeddings of individual words are computed by running the skip-gram model over the MAG corpus.

**twitch-gamers** (Lim et al., 2021) is a connected undirected graph of relationships between accounts on the streaming platform Twitch. Each node is a Twitch account, and edges exist between accounts that are mutual followers. The binary lassification task is to predict whether the channel has explicit content.

**pokec** (Lim et al., 2021) is an online social network, its task is to predict reported gender, certain account labels, or use of explicit content on user accounts, which is the same to twitch-gamers.

**snap-patents** (Lim et al., 2021) is a large-scale patent citation network dataset. Its primary goal is to predict the year a patent was granted, treating it as a node-level prediction task where each patent is a node and each citation is a directed edge.

**IMDB-BINARY**, **IMDB-MULTI**, and **REDDIT-BINARY** (Morris et al., 2020a) are benchmark datasets for graph-level classification. The IMDB datasets provide actor collaboration graphs for binary and multi-class movie genre prediction, respectively. REDDIT-BINARY consists of graphs from online discussion threads, with the task being to classify the community type.

### D.2    BASELINES

In our study, we mainly compare our SDE-GNN model with 15 baselines. The desciptions of these baselines are as following:

**GCN** (Kipf & Welling, 2017), **GAT** (Velickovic et al., 2018), **GraphSAGE** (Hamilton et al., 2017), and **GIN** (Xu et al., 2019) are classic graph neural network models that rely on a basic message-passing framework without explicit distance-aware mechanisms, which limits their expressive capacity.

**GraphGPS** (Rampásek et al., 2022) proposes a modular, scalable graph Transformer architecture with linear complexity by decoupling local edge-based message passing from global Transformer attention mechanisms, while maintaining universal function approximation capabilities.

**Exphormer** (Shirzad et al., 2023) constructs a graph Transformer architecture with linear complexity and superior theoretical properties by introducing a sparse attention mechanism based on virtual global nodes and expander graphs. With core components including expander graphs and virtual nodes, the computational complexity is reduced from the quadratic level of traditional graph Transformers to linear scaling with graph size.

**SGFormer** (Wu et al., 2023) addresses the scalability challenge of graph Transformers by proposing a simplified architecture that discards complex multi-layer multi-head attention structures. Instead, it leverages a single-layer attention mechanism capable of propagating information across arbitrary nodes with linear complexity, eliminating the need for positional encodings, feature/graph pre-processing, or augmented loss functions.

**GOAT** (Kong et al., 2023) introduces a scalable global Transformer framework that enables adaptive learning of homophily/heterophily relationships by allowing each node to attend to all nodes through an approximate, theoretically-justified global self-attention mechanism. This design eliminates domain-specific inductive biases while remaining compatible with both homophilic and heterophilic settings.

**ESAN** (Bevilacqua et al., 2022) enhances MPNN expressiveness by aggregating information from subgraphs (instead of direct node interactions), using predefined policies and equivariant architectures to distinguish MPNN-indistinguishable graphs. Theoretically grounded in novel 1D-WL test variants, it introduces stochastic subgraph sampling to reduce computational costs while maintaining flexibility in subgraph selection and design.

**Graphormer** (Ying et al., 2021a) addresses the challenge of adapting the powerful Transformer architecture for graph representation learning. It augments the standard Transformer with several novel structural encoding methods—such as Centrality Encoding, Spatial Encoding, and Edge Encoding—which are incorporated directly into the self-attention mechanism.

**GRIT** (Ma et al., 2023) introduces a graph Transformer that integrates inductive biases without message-passing, addressing prior limitations. Key innovations include: relative positional encodings initialized via random walk probabilities, an attention mechanism updating both node and node-pair representations, and degree injection in each layer. Theoretically, GRIT captures shortest-path distances and propagation matrices.

**SPE** (Huang et al., 2024) addresses Laplacian-based positional encoding limitations in graph Transformers by introducing a stable framework using a "soft partition" mechanism guided by eigenvalues. It resolves non-uniqueness and instability from hard eigenspace partitions, ensuring robustness to perturbations while maintaining universal expressiveness for symmetry-preserving functions.

**NeuralWalker** (Chen et al., 2024) bridges the gap between message-passing GNNs (which excel at local relations but fail at long-range dependencies) and graph Transformers (which oversimplify graphs via fixed-length vectors). It combines random walks (for long-range context) with local message passing by treating random walks as sequences and leveraging sequence models to capture dependencies.

**DE-GNN** (Li et al., 2020) introduces Distance Encoding (DE), a general class of structure-related features that measures the distance from a target set of nodes to every other node in the graph, using metrics like shortest path or PageRank. DE enhances GNNs by being used either as extra node features or as controllers of the message aggregation step, allowing the model to gain expressive power beyond the 1-WL test.

**RD-WL**. GD-WL (Zhang et al., 2023) overcomes the expressive limitation beyond the standard Weisfeiler-Lehman test by introducing a principled framework based on generalized graph distances, such as resistance distance (for RD-WL), to systematically enhance expressive power.

### D.3 HYPERPARAMETERS

For our SDE-GNN model, we employ grid search to identify optimal hyperparamters. The reduced dimension of dimenonality reduction is set to 1000 for two large-scale datasets—ogbn-arxiv and twitch-gamers. While for other medium- or small-scale datasets, the dimenonality reduction is not applicable. The power of adjacency matrix $k$ is searched within the range of 1 to 5. The maximum order of polynomial $m$ is searched within the range of 1 to 5. The number of layers $L$ is searched within the range of 1 to 10 (For CLUSTER and PATTERN datasets, the range is 1 to 40). The node embedding dimension is searched in $\{64, 128, 256, 512, 1024\}$. And the learning rate is searched in $\{10^{-4}, 10^{-3}, 10^{-2}, 10^{-1}\}$.

For the baseline models to compare with, we also employ grid search to optimize key hyperparamters and report the performance of the best configuration.

### D.4 EVALUATION SETTINGS

We uniformly employ classification accuracy on the test set to measure models' performance. Note that for CLUSTER and PATTERN dataset, since their node classes are imbalanced, we use the average node-level accuracy weighted with respect to the class sizes following (Dwivedi et al., 2023). The IMDB-BINARY, IMDB-MULTI, and REDDIT-BINARY datasets are randomly partitioned into training, validation, and test sets in a 5:2:3 ratio. While the data splits for the remaining datasets are kept consistent with those of the baseline methods (Dwivedi et al., 2023; Luo et al., 2024). Each model is run five times under identical conditions, with the mean and standard deviation of the results reported for statistical reliability. Evaluations are conducted on a machine with 192GB RAM, two 28-core Intel Xeon CPUs (2.2GHz), and an NVIDIA GeForce RTX 3090 GPU (24GB memory).

## E MORE EXPERIMENTAL RESULTS

### E.1 TIME AND MEMORY CONSUMPTION AT DIFFERENT NODE SCALES

To assess the scalability of different methods, we generate a series of synthetic graphs with node counts ranging from 200 to 200,000. The edges are randomly generated, maintaining a fixed 10:1 edge-to-node ratio. As illustrated in Figure 4, both the inference time and GPU memory footprint of our SDE-GNN model scale linearly with the number of nodes, empirically verifying the linear computation complexity of SDE-GNN.

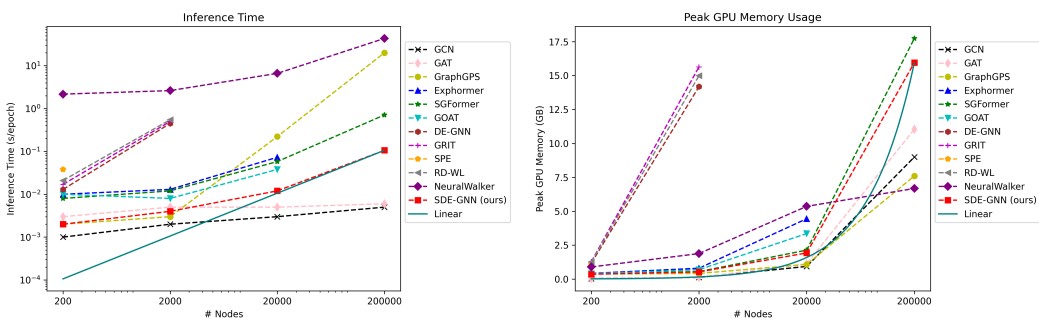

Figure 4: The comparison of models' inference time (s/epoch) and GPU memory usage (GB) at different node scales.

### E.2 ANALYSIS OF DIMENSIONALITY REDUCTION METHODS

We also investigate the impact of the selected dimensionality reduction method and the target dimension size on ogbn-arxiv dataset. As illustrated in Figure 5, the results show that 1) Randomized SVD (Halko et al., 2011) consistently outperforms other techniques, namely Count-Sketch (Charikar et al., 2002) and Gaussian Random Projection (Johnson et al., 1984), and 2) Model performance is positively correlated with the target dimensionality, while reducing the dimension from 1,000 to 200 does not lead to a significant drop in performance.

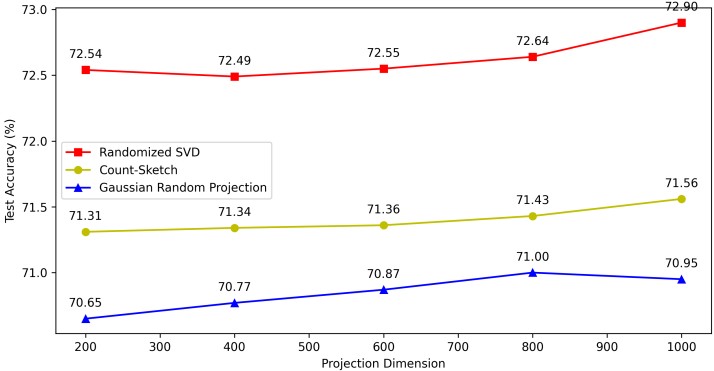

Figure 5: Comparison of different dimenionality reduction methods and reduced dimension.

## F LIMITATION

In this paper, we show that the adopted adjacency-power-based distance feature theoretically upper bounds the expressiveness of the eigenspace projection distance. However, whether this bound is strict or the two are equally expressive remains an open question, which we leave for future investigation.

## G USE OF LLMS

This article used LLMs as a tool to enhance writing clarity and correct grammatical errors. It is important to note that the fundamental ideas and the overall framework of this research are the original contributions of the authors.

