# OpenReview forum: "Towards Scalable Distance-Enhanced Graph Neural Network"
_ICLR.cc/2026/Conference — Submitted to ICLR 2026_

### Official Review · Reviewer_gsbM · 2025-10-28

**Soundness:** 2
**Presentation:** 3
**Contribution:** 3
**Rating:** 4
**Confidence:** 3

**Summary:**

In this paper, the authors proposed SDE-GNN, which uses the power of the adjacency matrix as general distance features to enhance the expressiveness of GNN. To reduce the computational cost, SDE-GNN leverages randomized SVD to project the matrix into a low-dimensional space for learning. The authors provide both theoretical results and practical performance. Overall, SDE-GNN achieve great results over baselines.

**Strengths:**

- The paper is well-written
- The authors provide extensive experiments to show the effectiveness of the methods. The experimental design is comprehensive and make sense.

**Weaknesses:**

- The major concern is that the SDE-GNN uses SVD on the adjacency matrix, which is sensitive to node permutation, which means that the learned distance encoding can be changed given a different permutation. This is, in general, counterintuitive and can bring potential learning issues, especially for graph-level tasks.
- The authors use randomized SVD to compute the distance features. I am wondering what the approximation ability of randomized SVD is compared to directly computing the order of the adjacency matrix, especially on higher orders. What's the practical time cost for this procedure? How well can the method be for large-scale graph, in term of both approximation ability and time conspution?
- The paper lacks experiments on some standard molecule datasets used to evaluate the practical expressiveness of GNN models like ZINC, QM9, or Peptides.

**Questions:**

See above.

---

### Official Review · Reviewer_ow5A · 2025-10-30

**Soundness:** 2
**Presentation:** 3
**Contribution:** 2
**Rating:** 4
**Confidence:** 4

**Summary:**

This paper addresses the scalability and expressiveness limitations of distance-enhanced graph neural networks (DE-GNNs). The authors propose a new model named Scalable Distance-Enhanced GNN (SDE-GNN). In this framework, pairwise adjacency-power-based distances are represented as the product of two asymmetric node encodings, enabling linear-time computation with respect to the number of edges. To improve efficiency, the authors adopt SVD for dimensionality reduction and exploit adjacency sparsity to compute the first-order term directly. The paper also provides a theoretical analysis showing that adjacency-power-based distances are more expressive than the eigenspace projection distance features. Empirical results show that SDE-GNN achieves strong performance, scalability, and efficiency compared to existing distance-based GNNs.

**Strengths:**

1. The paper introduces an efficient method to compute adjacency-power-based distance features, reducing the original quadratic complexity to linear in the number of edges.

2. The authors shows that the proposed adjacency-power-based distances are provably more expressive than eigenspace projection–based distance features.

3. Extensive experiments on 17 datasets demonstrate that the proposed SDE-GNN achieves better accuracy, scalability, and efficiency compared to existing distance-enhanced GNN baselines.

**Weaknesses:**

1. The adjacency-power-based distance leverages higher-order powers of the adjacency matrix, whereas the eigenspace projection–based distance only uses the Laplacian itself. Thus, it is somewhat natural that the former one achieves higher expressiveness since it contains richer structural information. However, once the randomized SVD is applied, the theoretical expressive power becomes unclear.

2. The adjacency-power-based distance has been explored in previous works (as noted in Remark 1 of the paper), and the main methodological novelty here lies in applying randomized SVD to accelerate its computation. While this reformulation improves scalability, the conceptual contribution beyond computational efficiency appears relatively limited.

3. The main concern lies in the ablation results (Table 5). The variant w/o polynomial keeps only the first-order term in Eq. (3), which does not require the proposed rSVD-based acceleration for higher-order terms. However, the accuracy gap between SDE-GNN and w/o polynomial is very small. Among four datasets, three of them show a difference below 0.15, which is within the reported standard deviation. This suggests that the additional polynomial terms may not provide a significant benefit.

4. Some design choices are not clearly explained. For instance, the motivation for replacing the sigmoid activation (line 184) with the learnable polynomial mapping in Eq. (3) is not clarified, and no experimental comparison with the sigmoid baseline is provided. Similarly, the substitution of $e$ with $V$ in Eq. (7) lacks justification. Originally $e$ corresponds to an identity matrix, and it is unclear why it is changed to $V$ in this formulation.

**Questions:**

1. The paper argues that adjacency-power-based distances are more expressive than eigenspace projection–based ones. However, once randomized SVD is applied for dimensionality reduction, does this expressiveness theoretically still hold? Could the authors clarify whether the low-rank approximation preserves or potentially weakens the theoretical guarantees?

2. In Table 5, the w/o polynomial variant (which keeps only the first-order term in Eq. (3)) performs comparably to the full SDE-GNN. Could the authors provide additional ablation results on more datasets to confirm whether higher-order terms consistently improve performance?

3. In Table 5, w/o polynomial achieves 65.91% accuracy on twitch-gamers, whereas in Table 6, when the polynomial expansion is used but the rSVD dimension p=100, the accuracy drops to 65.03%. This suggests that including higher-order terms may even degrade performance when p is not well tuned. Could the authors discuss this interaction between the polynomial expansion and the rSVD dimensionality?

4. Could the authors clarify the necessity of replacing the sigmoid activation (line 184) with the learnable polynomial mapping in Eq. (3)? Also, what is the motivation for replacing e with
V in Eq. (7)?

5. Minor issues (formatting and typos):
a) Figure 1 and the paragraph describing it appear on separate pages; please consider adjusting the layout for readability.
b) In line 217, the symbol e_v seems to be a typo and should likely be e_u.

---

### Official Review · Reviewer_WEoZ · 2025-11-01

**Soundness:** 3
**Presentation:** 2
**Contribution:** 2
**Rating:** 2
**Confidence:** 4

**Summary:**

The paper improves Distance-Encoding GNN by proposing a more scalable architecture that improves expressive power without incurring quadratic complexity. The key idea is to reformulate pairwise distance features using asymmetric node encodings and compress them with Randomized SVD, reducing the cost to near-linear. Experiments on classification tasks show that the proposed method achieves better accuracy, scalability, and efficiency compared to existing expressive GNNs.

**Strengths:**

1. The paper tackles a well-motivated problem which is to make expressive DE-GNNs more scalable.
2. There are some theoretical results that link adjacency-power-based distance and eigenspace projection distance.
3. Experiments show performance improvement on many datasets on node classification tasks,  though it critically lacks link prediction results.

**Weaknesses:**

1. The authors have submitted non-existing which hurts reproducibility: an error "The requested file is not found" occurs when clicking on the code url.
2. The experiment design has major flaws. A main advantage of DEGNN is its outstanding performance in link & node set prediction, as shown in the experiment of the original paper. However, this work has not done any experiment on link or node set prediction, but rather focuses on node & graph classification tasks.
3. While improving efficiency via Randomized SVD and decoupled message passing is interesting, the overall framework still largely builds on existing DE-GNN formulations and standard matrix approximation techniques. The contribution is incremental, and given 1 and 2, I think its current form is more publishable as a workshop paper.
4. There are multiple typos: “1-Weifeiler-Lehman” should be “1-Weisfeiler-Lehman”; “Existing methods can broadly divided” should be "can be broadly divided"; “demand higher expressive capacity” should be “demand a higher expressive capacity”. The paper needs to be further proofread.

**Questions:**

Please see weaknesses.

---

### Official Review · Reviewer_LbV8 · 2025-11-11

**Soundness:** 3
**Presentation:** 3
**Contribution:** 2
**Rating:** 4
**Confidence:** 5

**Summary:**

This paper proposes a Scalable Distance-Enhanced Graph Neural Network to address the quadratic complexity of traditional DE-GNNs. The core contribution is a distance-aware message-passing framework that uses adjacency-power-based distance features. To achieve scalability, the authors reformulate these features as a product of asymmetric node encodings and apply Randomized SVD for dimensionality reduction. This new formulation reduces the computational complexity to linear in the size of edges, making it applicable to large graphs. The authors also provide a theoretical analysis showing their chosen distance feature is highly expressive, upper-bounding other common metrics.

**Strengths:**

- The reformulation of distance-enhanced message passing, which allows for a scalable approximation using Randomized SVD, is new.
- SDE-GNN can run on large-scale graphs (e.g., pokec, snap-patents) where many competing expressive GNNs fail due to OOM errors.
- The SDE-GNN model's designs are well-justified and provide a theoretical insight into its chosen $d^{AP}$ feature, linking its expressive power to the Generalized Distance WL (GD-WL) test.

**Weaknesses:**

- The paper's focus on node/graph classification is limited. A highly similar idea, as seen in PEG (Wang, ICLR’22), utilizes decomposed structural features (Laplacian embeddings) to approximate distances and create edge weights to improve model expressiveness. PEG demonstrated great performance on link prediction tasks, whereas SDE-GNN was not examined on link-level tasks, which is a major motivation for more expressive power.
- The paper motivates its method with its expressiveness, but it should have included standard experiments, such as substructure counting, to empirically validate its power.
- The analysis of adjacency-power-based distance only covers the linear part of the model, leaving it unclear how the projection and non-linear polynomial expansion actually impact the theoretical expressive power.

Wang, Haorui, et al. "Equivariant and stable positional encoding for more powerful graph neural networks." ICLR’22.

**Questions:**

1) Can the authors please clarify the exact procedure for applying SDE-GNN to graph tasks under inductive settings? How are the encodings ($E$ and $e$) generated for a new, unseen graph at test time?
2) Why was the model not evaluated on link prediction (a natural fit for a distance-aware model) or substructure counting (a standard test for >1-WL expressiveness)?
3) Can the authors provide the end-to-end inference time comparison with baselines, including the wall clock time of SVD pre-computation, projection, and expansion, to more accurately assess the model's practical efficiency?
4) The theoretical analysis in Section 4 focuses on the $d^{AP}$ distance feature. Can the authors provide any analysis or intuition on how the dimension reduction and non-linear polynomial expansion affect the model's position in the GD-WL hierarchy?

---

### Meta-Review · Area_Chair_pHbC · 2026-01-07

**Summary:**

This paper improves Distance-Encoding GNN by proposing a more scalable architecture that improves expressive power without incurring quadratic complexity. The key idea is to use adjacency-power-based distance features to achieve scalability by reformulating these features as a product of asymmetric node encodings and apply Randomized SVD for dimensionality reduction, reducing the cost to near-linear  in the size of edges. Experiments on classification tasks show that the proposed method achieves better accuracy, scalability, and efficiency compared to existing expressive GNNs.

**Reviewer Concerns:**

The concerns are around limited novelty, limited evaluation task, lack of theoretical analysis of expressiveness, limited experimental validation, etc. The authors did not rebuttal. Some major concerns are summarized as follows:

[Reviewer WEoZ/ow5A] The novelty is incremental as the framework largely builds on existing DE-GNN formulations and standard matrix approximation techniques.

[Reviewer LbV8/ow5/gsbM] As the adjacency-power-based distance has been explored in previous works, the main novelty lies in applying randomized SVD to accelerate its computation. However, SDE-GNN uses SVD on the adjacency matrix, which is sensitive to node permutation. Once the randomized SVD is applied, the theoretical expressive power becomes unclear. The authors didn't provide any analysis or intuition on how the dimension reduction and non-linear polynomial expansion affect the model's position in the GD-WL hierarchy?

[Reviewer LbV8/WEoZ] A main advantage of DEGNN is its outstanding performance in link & node set prediction, while this paper only focuses on node/graph classification.

**Reviewer Scores:**

The authors didn't rebuttal.

---

### Decision · Program_Chairs · 2026-01-26

Reject